# Differences in Motor Imagery Ability between People with Parkinson’s Disease and Healthy Controls, and Its Relationship with Functionality, Independence and Quality of Life

**DOI:** 10.3390/healthcare11212898

**Published:** 2023-11-03

**Authors:** María del Rosario Ferreira-Sánchez, Marcos Moreno-Verdú, María de los Ángeles Atín-Arratibel, Patricia Martín-Casas

**Affiliations:** 1Department of Radiology, Rehabilitation and Physiotherapy, Faculty of Nursing, Physiotherapy and Podiatry, Complutense University of Madrid, 28015 Madrid, Spain; mrosario.ferreira@ucavila.es (M.d.R.F.-S.); matin@ucm.es (M.d.l.Á.A.-A.); pmcasas@ucm.es (P.M.-C.); 2Department of Physiotherapy, Catholic University of Avila, 05005 Avila, Spain; 3Department of Physical Therapy, Madrid Parkinson Association, 28011 Madrid, Spain; 4Faculty of Experimental Sciences, Francisco de Vitoria University, 28223 Pozuelo de Alarcón, Spain; 5Brain Injury and Movement Disorders Neurorehabilitation Group (GINDAT), Institute of Life Sciences, Francisco de Vitoria University, 28223 Pozuelo de Alarcón, Spain

**Keywords:** motor imagery, Parkinson’s Disease, vividness, temporal accuracy

## Abstract

Motor imagery (MI) has been shown to be effective for the acquisition of motor skills; however, it is still unknown whether similar benefits can be achieved in neurological patients. Previous findings of differences in MI ability between people with Parkinson’s disease (PwPD) and healthy controls (HCs) are mixed. This study examined differences in the ability to both create and maintain MI as well as investigating the relationship between the ability to create and maintain MI and motor function, independence and quality of life (QoL). A case–control study was conducted (31 PwPD and 31 HCs), collecting gender, age, dominance, socio-demographic data, duration and impact of the disease. MI intensity (MIQ-RS and KVIQ-34) and temporal accuracy of MI (imagined box and block test [iBBT], imagined timed stand and walk test [iTUG]) were assessed. Functional and clinical assessments included upper limb motor function, balance, gait, independence in activities of daily living and quality of life measures. Statistically significant differences in temporal accuracy were observed and partial and weak relationships were revealed between MI measures and functioning, independence and QoL. PwPD retain the ability to create MI, indicating the suitability of MI in this population. Temporal accuracy might be altered as a reflection of bradykinesia on the mentally simulated actions.

## 1. Introduction

Motor imagery (MI) is defined as the cognitive process of imagining executing an action, without actually performing any movement and without producing muscular tension [1]. This process involves the activation of the brain regions responsible for movement preparation and execution, as well as its voluntary inhibition through the activity of the primary motor cortex on the corticospinal pathway [1,2]. Depending on the sensory modality used during MI, motor images can be either visual or kinaesthetic [3]. Visual MI produces a visual representation of movement in which the subject is a spectator; he or she is able to observe the action in first or third person, in the form of an external image. This occurs when the person tries to see him/herself performing the movement [2,3].

Secondly, kinaesthetic MI is closely related to the proprioceptive experience associated with movement. The subject must create the representation by recalling the kinaesthetic sensations. In this case, the person is the performer; he/she performs and feels the movement in the first person, in the form of an internal image. This modality of MI achieves greater activation of sensorimotor areas and reaches electromyographic activity of the musculature involved [3,4].

MI has a multidimensional nature due to the complexity of the underlying processes. Previous studies confirm that motor imagery creation, maintenance and manipulation are independent skills, although they all contribute to performance [4,5]. In clinical practice and research, tools are available to assess each domain. Questionnaires such as KVIQ or MIQ-RS assess the ability to create the image, while mental chronometry tests allow for the assessment of maintenance [2,3]. In recent years, neuroimaging methods have also made it possible to describe more accurately the events occurring in cortical and subcortical regions during the imagination process; however, these are not always available in daily clinical practice [5].

Several systematic reviews have shown that the combined use of mental and physical practice is more effective than their isolated use, as MI training accelerates motor learning [5]. This type of training activates neuroplastic processes, which lead to improvements in performance in athletes [6] or the maintenance of the joint range and muscle strength after prolonged immobilization [7]. Despite its proven efficacy in healthy controls, it is still unknown whether similar benefits can be achieved in people with neurological conditions. Studies have observed changes and alterations in the ability to perform MI of people with stroke, multiple sclerosis and Parkinson’s disease (PD) [8], which could limit the application of this approach in neurorehabilitation settings.

PD is the second most common neurodegenerative disease in Western countries [9]. Cardinal motor symptoms of the disease are bradykinesia, tremor, rigidity and postural instability, accompanied by other non-motor symptoms such as olfactory dysfunction, psychiatric, cognitive, sensory symptoms or sleep disorders, which frequently are manifested prior to the formal diagnosis of the disease and result in a marked decrease in the quality of life [10,11]. The confluence of these symptoms produces alterations in the representation of the body with respect to space (body schema), which could limit the use of MI training in this population [8,12]. However, it is unknown whether there are specific clinical aspects of PD directly limiting the ability to perform MI, or whether this could limit the application of MI training.

The literature is mixed with regard to MI ability differences between people with PD and healthy controls. Whether differences in the ability to create MI are observed depends on the task assessed. Some studies have found differences in mental chronometry tasks, where people with PD show slowing of both imagination and execution [5,6,12]. These findings are supported by the strong evidence found on the correlation between the time needed to imagine an action and the time needed to execute it [6,13]. On the other hand, performance in mental rotation tasks and imagery vividness may not differ from healthy controls [14]. In contrast, other studies have found differences in these tasks in people with PD, secondary to alterations in fronto-striatal motor systems and parietal lobes, which are involved in integrating visuospatial information and imagery [15]. Also, factors such as the degree of severity of left side bradykinesia have been found to correlate with the vividness of MI, suggesting that specific motor symptoms of PD may alter these abilities [16]. 

Important discrepancies exist in the literature regarding the ability of people with PD to perform MI, and evidence of its relationship with clinical or functional variables is lacking. Therefore, our primary aim was to examine whether there are differences in the ability to create and maintain MI between people with PD and healthy individuals, using a range of measures. Our secondary goal was to study the relationship between the ability to create and maintain MI and functionality (balance, gait and upper limb motor function), the degree of independence in activities of daily living (ADL) and quality of life (QoL).

## 2. Materials and Methods

### 2.1. Study Design

An observational case–control study was conducted at a local Parkinson’s Association following the STROBE statement [17]. The study was approved by the Ethical Committee for Clinical Research of the Clínico San Carlos Hospital (19/166-E_Tesis), and written informed consent was obtained from each person prior to enrolment. All procedures were in accordance with the Declaration of Helsinki [18]. 

### 2.2. Participants

For the group of people with PD, the inclusion criteria were (a) diagnosis of Idiopathic PD, according to the United Kingdom Parkinson Disease Society Brain Bank Criteria [19]; (b) functional ability to perform all assessments (which include independent standing, autonomous gait and no significant range of motion limitations); and (c) age > 60 years. Exclusion criteria: (a) cognitive impairment (Mini-Mental State Examination < 24) [20]; and (b) diagnosis of neurological diseases other than PD, psychiatric diseases, orthopaedic or cardiovascular diseases, or presence of sensory deficits (visual and/or auditory) which could interfere with the assessments.

For healthy controls exclusion criteria were (a) history of neurological and/or psychiatric disease and (b) limitations of physical capacity according to the same criteria used for people with PD.

### 2.3. Assessments

Gender, age and hand dominance were collected as sociodemographic data, and disease duration, most affected hemibody, and MDS-UPDRS [21] and Hoehn and Yahr [22] scores as clinical data.

#### 2.3.1. Measures of Vividness: Ability to Generate MI

Spanish Version of the MIQ-RS questionnaire: It is a 7-movement self-administered instrument that assesses visual and kinaesthetic imagery ability [23]. Each movement is imagined using both sensory modalities and therefore the questionnaire has 14 items. Each item entails four steps: (1) adopting an initial position; (2) physically performing a movement; (3) returning to the initial position; and (4) visually or kinaesthetically imagining that movement. After the imagination of each movement, the person rates the ease or difficulty of generating that image on a 7-point scale from 1 = very hard to see/feel to 7 = very easy to see/feel. 

Spanish Version of the KVIQ-34 questionnaire [24]: It includes visual and kinaesthetic subscales, and the assessment process also comprises four steps. The KVIQ-34 uses ten simple movements of the neck, trunk and upper and lower limbs. Limb items are administered bilaterally. The person rates the intensity of the sensory information perceived on a 5-point scale, from 1 = no image/sensation to 5 = image as clear as actually seeing/feeling it.

Both questionnaires (MIQ-RS and KVIQ-34) assess MI vividness and have recently been validated for Spanish people with PD [25].

#### 2.3.2. Measures of Temporal Accuracy: Ability to Maintain MI

The imagined Timed Up and Go Test (iTUG) was used as a mental chronometry measure to assess the temporal accuracy of a lower limb action [26]. In this version, after the original TUG test, the participant mentally performs the same task, and the time needed is recorded. Outcomes of the iTUG were reported in seconds (TUG minus iTUG) and absolute percentage error (TUG minus iTUG divided by TUG), which represent the time discrepancy between real and mental tasks, scores ranging from 0 to +1 where scores near to zero mean better performance. The iTUG has been previously used in people with PD with satisfactory measurement properties [27].

An imagined version of the Box and Blocks Test (iBBT), which has been previously utilized in people with PD 13, was used as a measure of mental chronometry of upper limb tasks [28]. The original test assesses unilateral gross manual dexterity [29]. The person transfers 2.5 cm-sided cubes from one compartment of the box to the other, and the number of cubes transferred in 1 min is recorded as the outcome. In the iBBT, people had to physically transfer a total of 20 cubes and the time needed was recorded. The participants then had to perform the same task but mentally imagine it, and the time was also obtained. The test was performed with both hands (first with the dominant side). Outcome measures were obtained with the same formulae as the iTUG.

Functional and clinical assessment: The traditional BBT [29] and TUG [30] assessments were used to evaluate upper limb motor function and functional mobility, and the Berg Balance Scale [31] was used to assess static and dynamic balance. Cognitive assessment was performed using the Montreal Cognitive Assessment Scale (MoCA) [32]. The degree of independence in ADL was assessed using the Schwab and England scale [33] and QoL using the SF-36 questionnaire [34].

### 2.4. Procedure

People with PD were recruited from a local Parkinson Association. The source of controls was friends or relatives of participants with PD, through a 1:1 age and gender matching, to reduce biases introduced by these confounding factors. All participants were assessed with the paper-based Spanish KVIQ-34 and MIQ-RS questionnaires by the same experienced examiner at the Association facility. All participants were new to MI techniques. Due to the lengthy duration of the assessments, the MI and functional tests were administered in an interleaved form, allowing 5 min of rest between each test to avoid both mental and physical fatigue. Therefore, the test battery was administered in the following order: KVIQ—Berg Balance Scale—MIQ-RS—TUG—iTUG—BBT—iBBT—MDS-UPDRS—MoCA—Schwab and England—SF-36.

Participants were asked not to change their regular medication schedule and were evaluated in the “on” medication state (i.e., one to two hours after the anti-parkinsonian medication intake) [35]. Participants were excluded from the analyses in cases of missing data, due to not having correctly completed the tasks. 

### 2.5. Statistical Analysis

The data were analysed with the SPSS statistical package (version 29.0) (SPSS Inc., Chicago, IL, USA). Descriptive statistics was performed using a frequency distribution for qualitative variables and mean and standard deviation for quantitative variables. Statistical analyses were performed with alpha = 0.05 for statistical significance and 95% CI.

For between-group comparisons, the χ^2^ test was used for qualitative variables, the independent samples *t*-test was used for quantitative variables with normal distribution, and the Mann–Whitney U test was used as a non-parametric test. For within-group comparisons of different variables, the χ^2^ test was used for qualitative variables, and paired samples *t*-test or Wilcoxon rank test was used for quantitative variables. 

Correlation analyses were also undertaken between variables in both groups. Pearson’s or Spearman’s correlation coefficients were obtained according to the normal or non-normal distribution of the data.

## 3. Results

### 3.1. Participants

Sixty-two participants were recruited (31 people with PD and 31 healthy controls). Gender, age, dominance, MDS-UPDRS, Hoehn and Yahr stage, cognitive function, upper limb motor function, balance, gait, independence in ADL and quality of life are shown in Table 1 and revealed statistically significant differences between groups in upper limb motor function, balance, cognitive function, independence and quality of life.

### 3.2. Differences in the Ability to Generate MI: Vividness

There were no significant differences in imagery vividness between people with PD and healthy controls, in either for the total scores or the visual or kinaesthetic modalities, for both the KVIQ-34 and the MIQ-RS questionnaires, as shown in Table 2. 

No significant differences were found in the ability to create visual versus kinaesthetic MI in either the PD group (KVIQ-34 *p* = 0.709; MIQ-RS *p* = 0.208) or in healthy controls (KVIQ-34 *p* = 0.121; MIQ-RS *p* = 0.176).

The comparative analysis between the sides of the body revealed that there were no differences in the vividness of visual and kinaesthetic images created with the dominant side compared to the non-dominant side, either in people with PD (Visual *p* = 0.232; Kinaesthetic *p* = 0.665) or in healthy controls (Visual *p* = 0.924; Kinaesthetic *p* = 0.802), or between the more affected and the less affected sides (*p* = 0.163) in people with PD.

### 3.3. Differences in the Ability to Maintain MI: Temporal Accuracy

Performance on the mental chronometry tests is shown in Table 2. In the iBBT, there were statistically significant differences between people with PD and healthy controls (iBBT dominant side, seconds *p* = 0.014; iBBT non-dominant side, seconds *p* = 0.017), with healthy people performing better. Conversely, there were no significant between-group differences in the iTUG test (*p* = 0.632). Moreover, none of these differences were significant for the percentage outcome measure.

### 3.4. Relationship between MI Ability and Functionality, Independence and Quality of Life

Analysis of the relationships between the MI vividness measures and the upper limb motor function, balance and gait revealed no statistically significant correlations in people with PD (Appendix A). Also, no statistically significant correlations between vividness and impact and duration of the disease were found. On the other hand, statistically significant but weak correlations were found between the balance and gait test and iTUG in people with PD (Appendix A). 

The results revealed that MI vividness might not be related to motor function (Appendix A). Nevertheless, the relationship between temporal accuracy and functionality could be more relevant. Analysis showed a weak but significant relationship between balance and gait, and iTUG and iBBT of the non-dominant hemibody (Appendix A).

No significant relationships were found between ADL performance (Schwab and England scale) and the ability to create MI, neither in terms of vividness nor timing, neither in people with PD nor in healthy controls, as reflected in Appendix A. Nevertheless, a weak but statistically significant relationship between temporal accuracy (iTUG) and non-motor aspects of experiences of ADL (MDS-UPDRS Part I) was observed (Appendix A). No relationship was found between MI and the total cognitive and physical components of the SF-36 scale in people with PD; however, the physical component of QoL might be weakly related to MI vividness (KVIQ-34) and temporal accuracy (iTUG) in healthy controls (Appendix A).

## 4. Discussion

The main finding of this study is that there are no significant differences in MI vividness between people with PD and healthy controls, suggesting that the ability to generate MI is preserved in people with PD. The present finding refutes the theory that PD has an impact on the ability to create MI, even in mild to moderate stages, and suggests that basal ganglia impairment and body schema disturbance resulting from the confluence of motor and nonmotor symptoms may not limit the ability to create MI, as suggested in previous studies [8,12,36]. Furthermore, these results open a whole range of new therapeutic avenues. Our results indicate that people with PD have a preserved ability to create MI. Therefore, this demonstrates the validity and relevance of using MI-based therapeutic strategies in this population, which have been employed in the past. The present findings, together with the positive results on bradykinesia and other clinical manifestations of PD [37], argue that people with PD are good candidates for MI-based rehabilitation.

However, the temporal congruence of MI might be altered in people with PD. The ability to maintain the image during the upper limb task was significantly worse in people with PD; nevertheless, no difference was observed in the gait test. Previous evidence indicates that gait tasks and gait imagination [38] require greater involvement of semi-automatic and subcortical circuit-based control components [39], whereas upper limb tasks and imagination primarily involve higher cognitive, attentional and volitional functions [40]. The Hoehn and Yahr scale indicated that the postural control impairment of this sample of patients was mild–moderate; however, they were cognitively impaired compared to the control group. This could have led to the lack of differences in the performance and imagination of the gait task due to the relatively early stages of the disease, while the upper limb task did reveal differences between groups, due to the impairment of cognitive function and motor performance.

For both imagination tasks, the high standard deviation in the chronometry measure for the people with PD relative to the control group is noteworthy. This data reflects that performance was more variable in people with PD, and it hinders drawing definitive and reliable conclusions for temporal congruence between groups. In both cases, a factor that could have an impact on both prolonging and reducing imagination times was observed, and it is related to mental fatigue [41,42]. When fatigue appears, the subject can adopt two strategies: (a) try to be rigorous in the creation of the kinaesthetic image during the test and not reduce their levels of attention and involvement, which could be manifested by an increase in the time required to imagine the test with respect to the time used to perform it [41]; and (b) modify their patterns of action and imagination, with the desire to finish the test quickly and create a quick visual image, which is less precise [43,44]. This second strategy results in very reduced imagination times, sometimes reaching the implausible [45,46,47]. The first strategy was observed in the PD group, prolonging imagination times, and this is consistent with the literature, which suggests that this strategy could be a manifestation of bradykinesia on MI [5,6,8,12,48,49].

### 4.1. Functional Implications and Relationship with ADL and QoL

The present study did not find moderate or strong correlations between the ability to create MI and upper limb motor function, gait and balance, suggesting that people with PD retain movement representation, but execution components hinder functionality. Also, it is worth mentioning that there are large individual differences in MI even in the general population and these might also explain a null effect at this point, suggesting that MI might not be suitable for people unable to create vivid images [16,48].

Despite the non-significant relationship between the ability to create MI and functionality, possibly due to the intrinsic limitations of the study, there is evidence that MI-based interventions may improve gait speed, upper limb motor function and balance, following MI training [50,51,52,53,54]. This suggests that the ability to create MI and functionality might be related, even though not observed in this cross-sectional study. However, factors other than MI may explain differences in motor function, and further studies are needed to clarify these issues. Despite this, another line of work has found no relationship between gait and gait MI, suggesting that a mobility deficit does not impede task imagination [55]. In the case of balance, other studies point to independence between the two variables, given that balance responses are largely automatic, and it is difficult to be influenced by voluntary or cognitive components [56]. Therefore, this matter is currently under debate due to there being causal evidence between MI training on functionality, but not observational evidence between imagery ability and functionality.

It is important to notice that the statistically significant correlations between mental chronometry and motor function were only found in healthy people, probably because of the altered temporal accuracy of the MI in people with PD.

The present study did not find significant correlations between the ability to create MI and the degree of independence in ADL or QoL in people with PD, which suggests that they are independent constructs. Several systematic reviews report significant improvements in general mobility, postural stability, gait and motor symptoms after different interventions (physiotherapy, dance, aerobic training, aquatic gymnastics, etc.), which have been shown to have an impact on the quality of life of people with PD [57,58,59,60,61]. Thus, the ability to create MI may not be intrinsically associated with the degree of independence in ADL or QoL; however, this does not imply that MI training may not improve these variables by means of enhancing performance on functional tasks, as the extent to what MI interventions may or may not impact on the clinical or functional status of people with PD is still uncertain [56,62].

Previous studies confirm that patients with PD retain MI vividness, but this does not seem to correlate with performance in the gait task [63]. Neuroimaging techniques have allowed observing different patterns of brain activation during imagined walking, as PD patients have reduced activity in globus pallidus and increased activity in the supplementary motor area [63]. Likewise, previous research reports an increased activity in the right extrastriate body area and occipital–parietal areas, suggesting increased visual processing in order to compensate for a reduced MI ability and proprioceptive impairment in PD [64,65]. Specifically, in gait imagery, people with PD exhibited significantly greater activation in the left parietal operculum, left supplementary motor area and right cerebellum [66], and the characteristics of the imagined task (motor, dual, changing, multitasking, etc.) also lead to different activation patterns in PD patients [67]. 

Future research could further investigate the relationship between cortical and subcortical activation patterns in tasks of different characteristics, both in executed and imagined modality in PD patients. Also, further studies are required to fully investigate the relationship between the ability to maintain MI and independence and QoL, given that different correlations have been observed with two different rating scales (Schwab and England versus MDS-UPDRS Part I), or a relationship has only been found in healthy people (physical component of SF-36).

### 4.2. Limitations

Some limitations of this study are related to the small sample size (especially regarding the power of correlations), lack of generalizability based on the Hoehn and Yahr stage, and to the tests administered, especially MI assessments that require long imagery times, which might have produced mental fatigue and possible errors during test performance. Also, possible alternative measurements in MI such as neuroimaging tests using functional magnetic resonance imaging were not available.

## 5. Conclusions

People with PD retain the ability to create MI in terms of vividness; nevertheless, temporal accuracy might be altered as a reflection of bradykinesia on the mental images. The relationship between the ability to create MI and functionality, independence in ADL and quality of life in people with PD requires further investigation.

## Figures and Tables

**Table 1 healthcare-11-02898-t001:** Participants characteristics.

	PD Group (n = 31)		HC Group (n = 31)		*p*-Value
	Mean	SD	Mean	SD	
Age, years	73.61	7.08	73.32	8.16	0.583
Disease duration, years	9.13	6.12	NA	NA	-
MDS-UPDRS Total Score	57.77	17.29	NA	NA	-
MDS-UPDRS Part I	12.74	6.39	NA	NA	-
MDS-UPDRS Part II	15.71	6.46	NA	NA	-
MDS-UPDRS Part III	25.32	7.76	NA	NA	-
MDS-UPDRS Part IV	4.01	3.35	NA	NA	-
Berg Balance Scale	48.9	8.14	54.68	1.96	<0.001 *
Timed Up and Go Test	10.13	4.59	8.47	2.86	0.093
Box and Blocks Test, dominant side	37.32	8.31	47.9	11.03	0.011 *
Box and Blocks Test, non-dominant side	35.29	8.93	46.13	11.55	0.143
Box and Blocks Test, most affected side	35.71	9.11	NA	NA	-
Box and Blocks Test, less affected side	36.9	8.19	NA	NA	-
SF-36 Physical Scale, Total	46.05	18.05	78.55	10.11	<0.001 *
SF-36 Mental Scale, Total	56.97	19.14	75.96	17.67	<0.001 *
SF-36 Physical Functioning	50.65	26.26	87.42	13.9	<0.001 *
SF-36 Role Physical	47.58	39.45	92.74	17.31	<0.001 *
SF-36 Bodily Pain	54.35	22.56	82.58	15.61	<0.001 *
SF-36 General Health	43.06	14.47	66.61	16.3	<0.001 *
SF-36 Vitality	48.87	14.82	67.74	16.27	<0.001 *
SF-36 Social Functioning	63.31	23.48	85.48	19.66	<0.001 *
SF-36 Role Emotional	53.77	46.09	86.02	30.76	0.005 *
SF-36 Mental Health	62.84	16.56	72.65	19.27	0.032 *
MoCA	24.35	2.69	27.1	2.05	<0.001 *
	n	%	n	%	-
Gender, females (%)	13	41.9	13	41.9	1.000
Dominant side, right (%)	31	100	31	100	1.000
Hoehn and Yahr Stage 1, n (%)	1	3.2	NA	NA	-
Hoehn and Yahr Stage 1.5, n (%)	5	16.1	NA	NA	-
Hoehn and Yahr Stage 2, n (%)	5	16.1	NA	NA	-
Hoehn and Yahr Stage 2.5, n (%)	8	25.8	NA	NA	-
Hoehn and Yahr Stage 3, n (%)	12	38.7	NA	NA	-
Schwab and England Scale 40%, n (%)	1	3.2	-	-	-
Schwab and England Scale 60%, n (%)	2	6.5	-	-	-
Schwab and England Scale 70%, n (%)	3	9.7	-	-	-
Schwab and England Scale 80%, n (%)	10	32.3	1	3.2	-
Schwab and England Scale 90%, n (%)	14	45.2	-	-	-
Schwab and England Scale 100%, n (%)	1	3.2	30	96.8	-
Most affected side, right (%)	11	35.4	NA	NA	-
Most affected side, left (%)	15	48.4	NA	NA	-
Most affected side, bilateral (%)	5	16.2	NA	NA	-

NA: not applicable; HCs: healthy controls; MoCA: Montreal Cognitive Assessment; PD: Parkinson’s Disease; SF-36: Short Form Health Survey-36; MDS-UPDRS: Unified Parkinson’s Disease Rating Scale; *: statistically significant differences.

**Table 2 healthcare-11-02898-t002:** Motor imagery ability measures.

	PD Group (n = 31)		HC Group (n = 31)		*p*-Value
	Mean	SD	Mean	SD	
MIQ-RS Total	72.77	19.97	77.19	22.91	0.106
MIQ-RS Visual Subscale	36.84	11.62	39.29	12.6	0.090
MIQ-RS Kinaesthetic Subscale	35.94	10.65	37.90	11.64	0.203
KVIQ-34 Total	123.03	34.95	126.65	34.22	0.682
KVIQ-34 Visual Subscale, Total	61.35	21.46	65.65	21.74	0.283
KVIQ-34 Kinaesthetic Subscale, Total	61.68	17.7	61.00	18.87	0.811
KVIQ-34 Visual Subscale, dominant side	25.35	8.85	27.39	9.09	0.171
KVIQ-34 Visual Subscale, non-dominant side	25.87	9.04	27.23	9.29	0.363
KVIQ-34 Visual Subscale, most affected side	25.90	8.93	NA	NA	-
KVIQ-34 Visual Subscale, less affected side	25.32	8.96	NA	NA	-
KVIQ-34 Kinaesthetic Subscale, dominant side	25.84	7.59	25.13	8.08	0.777
KVIQ-34 Kinaesthetic Subscale, non-dominant side	25.68	7.25	25.03	8.1	0.767
KVIQ-34 Kinaesthetic Subscale, most affected side	25.94	7.5	NA	NA	-
KVIQ-34 Kinaesthetic Subscale, less affected side	25.58	7.33	NA	NA	-
iTUG, seconds	3.67	3.05	3.34	2.24	0.632
iBBT dominant side, seconds	6.15	4.22	3.85	2.79	0.014 *
iBBT non-dominant side, seconds	6.93	5.59	4.15	3.18	0.017 *
iBBT most affected side, seconds	6.54	4.75	NA	NA	-
iBBT less affected side, seconds	6.21	4.84	NA	NA	-
iTUG, percentage	35.58	20.07	37.67	18.01	0.667
iBBT dominant side, percentage	21.59	14.25	17.67	12.65	0.257
iBBT non-dominant side, percentage	23.05	16.44	16.49	11.67	0.087
iBBT most affected side, percentage	22.45	16.19	NA	NA	-
iBBT less affected side, percentage	21.22	13.83	NA	NA	-

NA: not applicable; HCs: healthy controls; iBBT: Imagined Box and Blocks Test; iTUG: Imagined Timed Up and Go Test; KVIQ-34: Kinesthetic and Visual Imagery Questionnaire, extended version; MIQ-RS: Movement Imagery Questionnaire, revised second version; PD: Parkinson’s Disease; *: statistically significant differences.

## Data Availability

Data are contained within the article.

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
