# Peer review of "Differences in Motor Imagery Ability between People with Parkinson’s Disease and Healthy Controls, and Its Relationship with Functionality, Independence and Quality of Life"

_healthcare, 2023, doi:10.3390/healthcare11212898_

Round 1
Reviewer 1 Report
Comments and Suggestions for Authors
The topic of motor imagery in Parkinson's disease patients is indeed very interesting. However, it is important to clearly distinguish the neurological networks and mechanisms involved in fine motor and gross motor skills. The research title and main text should explicitly outline this distinction and introduce relevant previous studies. It is necessary to re-evaluate whether the measurement tools used in this study accurately assess the research objectives and scope. The discussion should also be revised based on the evidence from clinical studies on the neural network mechanisms of motor imagery in PD patients.
Comments on the Quality of English LanguageMinor editing of English language required
Author Response
Dear Reviewer:
Thank you very much for your constructive comments to improve our manuscript. In the attached file you can check our responses to your comments and the changes made to the manuscript are highlighted in underline.
Kind regards.

Reviewer 2 Report
Comments and Suggestions for Authors
The introduction is too tight. MI should be explained more extensively and it should be explained clearly how it is measured.Is there way of measuring it with functional MRI or other functional techinques? if so why have they not been tested?
Author Response

(The authors gave the same response as above.)

Reviewer 3 Report
Comments and Suggestions for Authors
This paper reports on an observational case-control study of differences between people with PD and healthy age- and gender-matched controls in the ability to create and maintain mental imagery and relationship to function (gait, balance and upper limb), independence in activities of daily living, and quality of life. Using two validated and reliable measures of mental imagery vividness (MIQ-RS and KVIQ-34) and two measures of mental imagery temporal accuracy (iTUG for gait & balance and iBBT for upper limb function), the authors showed that participants with PD maintained the ability for vivid mental imagery, but with less temporal accuracy than for healthy controls, most notably for the upper limb task under greater voluntary/cortical control than the gait task under a degree of automatic/subcortical control. The authors also suggest these temporal delays may be related to mental fatigue. The authors conclude that the lack of correlation between the ability to create mental imagery and ADL independence or QoL suggests these are independent constructs yet doesn’t preclude that mental imagery can improve ADL independence and QoL through enhancing performance on functional tasks.
Major Strengths of the paper:
· - Work extends current understanding of mental imagery ability in people with PD, uniquely focusing on imagery of functional gait, balance and upper limb tasks and the relationship with ADL independence and QoL.
· - Design is appropriate and technically sound, analysis and results are clearly reported, and interpretation is fitting and supported by their results.
· - Paper is of substantial and important interest to readers, particularly clinicians working in rehabilitation with people with PD.
Minor Recommendations:
· - In suggesting that mental fatigue may have played a role in the temporal accuracy results, the authors should provide greater detail of the methods/procedure participants followed, such as timing of the actual and imagined tasks (any rest or alternate activity between actual and imagined performance).
· - Table 1 indicates scores for Trunk Impairment Scale, but this measure was not listed or described in the Methods, nor was it addressed in Results
· - Table 1 line for values for “Dominant side, right (%)” is left blank.
Comments on the Quality of English LanguageMinor grammatical errors should be corrected before publication.
Author Response

(The authors gave the same response as above.)

Round 2
Reviewer 2 Report
Comments and Suggestions for Authors
the paper is fine, now